

# Interaction of guidance types and the Need for Cognitive Closure in wiki-based learning

Sven Heimbuch and Daniel Bodemer

Media-Based Knowledge Construction, University of Duisburg-Essen, Duisburg, Germany

## ABSTRACT

One purpose of wikis is the collaborative generation of content. During creation processes, controversies between authors emerge that they discuss on the article's talk page. Research suggests that controversies based on opposing points of view and contradictory evidence can be fruitful to trigger individual elaboration processes. However, previous research also showed that many wikis are not necessarily suited to identify relevant discussion contents and thus users need additional support as guidance. In an experimental laboratory study ($N = 181$) on wiki talk pages, we investigated two guidance measures in conjunction with the need for cognitive closure: (1) visual markers to highlight controversy status (*implicit guidance*) and (2) a collaboration script that directs users towards discussions (*explicit guidance*). Effects on wiki processes and learning outcomes were analysed. The results show that both guidance types can affect user behaviours, but in interaction with the individual Need for Cognitive Closure there were no meaningful effects. With respect to learning outcomes, we found an anticipated pattern for the interaction of the Need for Cognitive Closure with both guidance principles. The data provides support for differences in the learning success depending on the provided guidance type and the individual Need for Cognitive Closure.

## GENERAL INTRODUCTION

To date, wikis are widely used in many educational contexts for collaborative knowledge construction and learning tasks (*Notari et al., 2016*). They can be suitable for the purpose of co-constructing knowledge artefacts and promoting learning processes, although the design of wikis is not immediately conducive to learning (*Capdeferro & Romero, 2012*). As such, wikis have already been used in various educational settings and have been integrated in many applications and assignments used for teaching (*Bartelsen & Brauer, 2010*). Especially in higher education, they can be implemented in almost all kinds of degree course programmes, to facilitate collaborative learning of new definitions and concepts. In comparison to other knowledge building platforms that have been deployed in educational contexts (*Scardamalia & Bereiter, 2006*), wikis enable their users to perform very influential and drastic changes to the whole environment and its shared artefacts (*Kimmerle et al., 2015*). From a constructivist's perspective, wikis inherently

Corresponding author
Sven Heimbuch,
sven.heimbuch@uni-due.de

have great potentials for collaborative learning as people learn better when they design the materials by themselves (*Cole, 2009*). During the collaborative co-construction of knowledge, opinion controversies and socio-cognitive conflicts can arise. Such conflicts emerge when a person's cognitive schemes are in contradiction with newly confronted schemes. Consequently, this leads to reorganisation and restructuring of cognitive processes, if consensus building is requested or required (*Bell, Grossen & Perret-Clermont, 1985*). These socio-cognitive conflicts are of particular significance for many learning-related fields such as Computer-Supported Collaborative Learning (CSCL). Collaborating in a group can lead to higher cognitive achievements compared to an individual working alone (*Doise, Mugny & Perret-Clermont, 1975*). Regarding wikis, opinion controversies and conflicts are more difficult to identify and to process because of the specific structure of wikis. We have previously shown in several experiments that additional implicit or explicit guidance in wikis can be beneficial for individual and collaborative learning processes and outcomes (*Heimbuch, Uhde & Bodemer, 2014*; *Heimbuch & Bodemer, 2015b*; *Heimbuch & Bodemer, 2017*). The research presented here builds and extends upon this by further investigating effects of a relevant individual variable, namely the Need for Cognitive Closure, and its interaction with different types of guidance.

## THEORETICAL AND EMPIRICAL BACKGROUND

Despite the acceptance of wiki usage for educational purposes, the effectiveness and efficiency for learning per se remains questionable due to ambiguous results in research. Fundamental processes of co-constructing socially shared artefacts within wikis are the internalisation and externalisation of knowledge from an individual's cognitive system into the wiki as a social system or vice versa (*Cress & Kimmerle, 2008*). In these processes lie potentials between collaborators for controversies to arise. Possible grounds for such controversies are different opinions or contradictory knowledge that can constructively foster learning outcomes (*Johnson et al., 1985*). Furthermore, content-related controversies offer opportunities to induce socio-cognitive conflicts that can trigger equilibration and elaboration processes and thus individual learning (*Mugny & Doise, 1978*). Since such conflicts do not have to be detrimental for successful learning, they can provide opportunities for constructive controversy resolutions. On existing wiki talk pages, a bandwidth of different conflict types can be found, ranging from socio-emotionally driven disputes to significant evidence-led discussions which comprise hidden potential for knowledge construction processes. Highlighting the latter kind of controversies in wikis' underlying discussion threads might guide interested individuals towards essential learning processes based on socio-cognitive conflicts. It is important to note that socio-cognitive conflicts by means of the co-evolution of knowledge model do not inevitably require that individuals must be involved in constant interaction with each other (*Cress & Kimmerle, 2008*). Even simple interactions of an individual's cognitive system with pre-existing contents in a social system that others have generated in a socially shared manner are socio-cognitive in that model. This kind of socio-cognitive conflict becomes especially clear in asynchronous systems such as wikis where no contributor has a guarantee

to receive direct or indirect feedback by others within a narrow timeframe or even at all (*Heimbuch & Bodemer, 2017*). Due to the large information mass that can be present on established wiki talk pages, it is also evident that users can easily be overwhelmed and might be unable to assess a source's quality that is involved in a controversy or is causing a socio-cognitive conflict.

Providing learners with media and letting them freely collaborate does not automatically promote systematic learning processes and is dependent on an interplay of numerous variables such as the task itself, characteristics of the group and its individual members or the underlying collaboration media (*Stahl, 2006*). It has been shown that missing objectives and a lack of structure are problematic for productive interactions and outcomes in a collaborative setting (*Bromme, Hesse & Spada, 2005*). Thus, guiding structured learning and communication processes is essential for the effectiveness of computer-supported collaborative learning settings (*Fischer et al., 2013*) and a certain level of coercion in the knowledge construction process is recommended to produce meaningful outcomes (*Papadopoulos, Demetriadis & Weinberger, 2013*). Users as wiki group members seek information on what is known by others for developing awareness of who knows what (*Noroozi et al., 2013*). With increasing complexity, further assistance in dealing with controversial information can become necessary. Cognitive Group Awareness (CGA) tools can be useful to provide beneficial assistance to the learner. These are tools with a focus on gathering and visualising knowledge-related contextual cues (*Bodemer & Dehler, 2011*). Concretely in the case of wikis, this can be achieved with minimal invasive modifications for wiki talk pages that make controversial discussions and their concurrent state of discussion progress more salient by adding visual highlights (*Heimbuch & Bodemer, 2017*). Another line of wiki-related research has proposed additional measures of explicit guidance to incorporate in wiki-based learning environments to improve the overall quality of knowledge artefacts and for better coordination processes of students. The implementation of collaboration scripts is one possible explicit guidance measure where the activities of writers and editors within a social system are coordinated and optimised (*Dillenbourg, 2002*). Positive effects have been found for scripts with a special focus on article editing and revising (*Wichmann & Rummel, 2013*) and collaboration scripts that set the focus on a priori discussion (*Heimbuch, Uhde & Bodemer, 2014*) that ultimately led to more coherent articles and fewer inaccurate articles. The latter script was aimed to engage participants to discuss any planned article edits and revisions upfront before changes to a document will be performed, resulting in a script called "Discuss, Deliberate, Revise" (DDR). This paper builds upon the research on the DDR collaboration script and takes it further by explicitly addressing relevant individual differences.

If a person is advised to work collaboratively in a more structured and coercive environment, there are indications that the effort a learner is willing to invest in searching for solutions to a problem can be influenced by the Need for Cognitive Closure (NCC). The NCC is a motivational continuum between the desire to acquire a clear answer in an ambiguous situation and the avoidance of quick and unambiguous answers. Various empirical results and discussions illustrate that it can be regarded as a relevant construct in knowledge creation processes (*Webster & Kruglanski, 1994*; *Heimbuch & Bodemer, 2017*).

Individuals with a high NCC want a definite answer in a judgement situation (*Schlink, 2009*). They are more likely to experience the need for reaching cognitive closure as quickly as possible and to try to maintain a state of achieved cognitive closure for as long as possible. People who score high on the constructs' scales tend to base their decisions on simple heuristics (*Dreu, Koole & Oldersma, 1999*), whereas low scoring individuals consult more information in situations of uncertainty (*Schlink, 2009*). Thus, we expect that in wiki-based learning individuals with a low need for cognitive closure are more likely to search purposefully for additional in-depth information about a topic in an ambiguous situation. Although there are close ties between the Need for Cognitive Closure and inter-individual differences in learning and knowledge construction, there are few studies in technology-enhanced learning to address this construct (*DeBacker & Crowson, 2009*).

## Research questions and hypotheses

*RQ1*: How do measures of (1) implicit guidance and (2) explicit guidance affect processes and outcomes in wiki groups?

Building upon the positive results of our previous experimental studies, distinct kinds of implicit and explicit guidance implementations for wiki-based learning environment have already been analysed (*Heimbuch & Bodemer, 2014*; *Heimbuch, Uhde & Bodemer, 2014*; *Heimbuch & Bodemer, 2015a*; *Heimbuch & Bodemer, 2015c*; *Heimbuch & Bodemer, 2016*; *Heimbuch, Ollesch & Bodemer, 2016*; *Heimbuch & Bodemer, 2017*). Both kinds of guidance (implicit and explicit) showed several positive effects on knowledge test scores and wiki contribution quality as well as potentials for more purposeful interactions within the wiki environment. On the one hand, making controversial discussions more salient to wiki users can be achieved by implicitly guiding learners towards those contents. Contro-versies between wiki editors can be quickly recognised by a user and this can furthermore lead to socio-cognitive conflicts between the user and the system. On the other hand, a more explicit collaboration script proposal aims at fostering deeper elaboration processes by encouraging discussions prior to the externalisation of knowledge into the wiki and has already shown to be beneficial for mentally integrating different perspectives on a topic.

*H1a*: The individual processing of wiki pages in terms of selecting topics and replying to discussions is expected to be equivalent for both wiki groups.

*H1b*: The individual contribution time is expected to be different between the guided wiki groups. Due to the nature of explicit guidance, it is expected that participants in this group spent more time on contributing to discussions in comparison to participants in the implicit guidance group.

*H1c*: The individual learning success is expected to be equivalent for both wiki groups, because neither guidance type is per se better than the other.

*RQ2*: How does the individual Need for Cognitive Closure influence processes and outcome variables related to learning in the case of (1) implicit guidance and (2) explicit guidance?

In our previous research in this area we have provided several indications that the Need for Cognitive Closure might have an impact on learning-related processes and outcomes (*Heimbuch & Bodemer, 2014*; *Heimbuch, Uhde & Bodemer, 2014*; *Heimbuch*

& Bodemer, 2016; Heimbuch, Ollesch & Bodemer, 2016; Heimbuch & Bodemer, 2017). To further support and extend on these findings a follow-up laboratory study was conducted to compare the effects of one implicit and one explicit guidance implementation for wiki environments in interaction with the individual Need for Cognitive Closure of learners.

*H2a*: The interaction of the Need for Cognitive Closure and the implemented guidance in the wikis determines the individual selection and reply behaviour. It is expected that high NCC participants select and reply mostly to resolved controversies when their status is visualised as it is the case in the implicit guidance wiki. For low NCC participants, it is expected that they behave equivalently in both wikis.

*H2b*: The interaction of the Need for Cognitive Closure and the implemented guidance in the wikis determines the individual contribution time. It is expected that high NCC participants spent more time contributing when implicit guidance is present. For low NCC participants, it is expected that they spent more time with contributions when explicit guidance is present.

*H2c*: The interaction of the Need for Cognitive Closure and the form of structuring determines the individual learning success. It is expected that high NCC participants score higher in a knowledge test when implicit guidance was provided. For low NCC participants, it is expected that they achiever higher scores when explicit guidance was present.

## METHOD

### Design and participants

The presented study was approved by the Ethics Committee of the Department of Computer Science and Applied Cognitive Science (INKO) at the University of Duisburg–Essen, Germany (IRB approval: psychmeth_2015_WIKI_08). We decided to use an experimentally controlled laboratory setting with individual participants. This decision has been made to isolate potential effects of the experiment's guidance types in conjunction with the individual Need for Cognitive Closure from interfering effects caused by social interactions that naturally occur in wiki environments. Consequently, a between-subjects design was used to investigate the interplay between different guidance types and the Need for Cognitive Closure in wiki-based learning. The first independent factor was the type of provided talk page guidance (implicit vs. explicit). The second factor of interest was the individual Need for Cognitive Closure, which was factorised via median splits into two levels (low vs. high). For deeper inferential analyses, linear regression models were specified to make use of the full interval data spectrum. Before conducting the experiment, two a priori power analyses were performed corresponding to the main hypotheses. The first power analysis for between-group equivalence was performed with the R package *TOSTER* (*Lakens, 2017a*) for equivalence hypothesis testing with parameters $\alpha = .05$ $1 - \beta = .90$, $d = [-0.5, 0.5]$. The bounds of $d = -0.5$ and $0.5$ were chosen, because they translated into a raw score difference of approximately 1 point in the knowledge test, which was considered as the smallest effect size of interest (SESOI). The second power analysis for interactions of the grouping condition with the NCC was performed for linear regression designs in G*Power (*Faul et al., 2007*) with parameters $\alpha$

$= .05, 1 - \beta = .90, f = 0.25$. As a result, the analyses suggested an optimal $N = 174$ for the equivalence hypothesis tests and $N = 171$ for potential interaction effects in a hierarchical linear regression model.

The recruitment took place in the Facebook groups and student forums of the University of Duisburg–Essen (Germany). Most of the participants were students recruited from the university's Applied Cognitive and Media Science degree program ($n = 168$; 92.82%). From the remaining $n = 13$ participants, $n = 11$ were students recruited on campus. We did not document their degree programmes in detail and subsumed them under the category "other". They were equally distributed over both experimental groups ($n = 5$ and $n = 6$). There was one participant in each experimental group who was not a student. After approximately five weeks, the recruitment of participants was terminated when no more subjects were willing to volunteer in the experiment. Finally, it was possible to sample $N = 181$ subjects with complete data sets. This sample size provides sensitivity for minimum equivalence bounds of $d = [-0.35, 0.35]$, which is more than sufficient for the study's planned bounds of $d = [-0.5, 0.5]$. The participants' age range was between 17 and 33 years ($M = 20.59, SD = 2.59$; $n_f = 136$ female, $n_m = 45$ male). Participants were randomly assigned on their arrival at the laboratory to one of the two learning environments, resulting in an equal distribution to both an implicit guidance wiki ($n_{imp} = 91$) and an explicit guidance wiki ($n_{exp} = 90$). The participants' overall topic-specific interest in the subject "forms of energy" was on a medium level ($M = 7.35, SD = 3.20$) and their self-assessed prior knowledge about the subject matter was relatively low ($M = 3.78, SD = 2.59$), on scales both ranging from "0 = low" to "15 = high". Differences between both wiki groups regarding topic-specific interest were very small, $U = 3,917.50$, $p = .614, d = .04$, 95% CI $[-.13, .21]$, $BF_{01} = 5.28$. Regarding prior knowledge, a small but meaningful difference between group is suggested by the data, $U = 3,359.50, p = .033, d = .18$, 95% CI $[-.02-.34]$, $BF_{10} = 2.16$. This difference in prior knowledge will be controlled for in respective analyses of the learning outcomes.

## Materials and wiki environments

Participants were confronted with different forms of energy, such as fossil fuels, nuclear power and renewable energy as the experiment's subject area. A base article on the topic was provided as an initial start page to provide a common ground for all participants in the experimental wikis. This article was derived from original sections of the German Wikipedia, and we adapted them for the study's purpose, resulting in an article with a total length of 630 words. From original talk page discussions on the corresponding Wikipedia articles, a total number of 12 discussion threads were generated with the aim of reproducing a wiki-like environment (Fig. 1). All discussion threads in the wiki were made up of at least two discussants. They were included on the experimental wiki talk page to represent existing discussions that directly relate to the main article. Six of the integrated discussions comprised content-related controversies with opposing points of views about forms of energy and were resolved in consensus after the exchange of a few evidence-based arguments. The other six discussion threads were open and unresolved controversial discussions where discussants did not reach any kind of consensus. Since

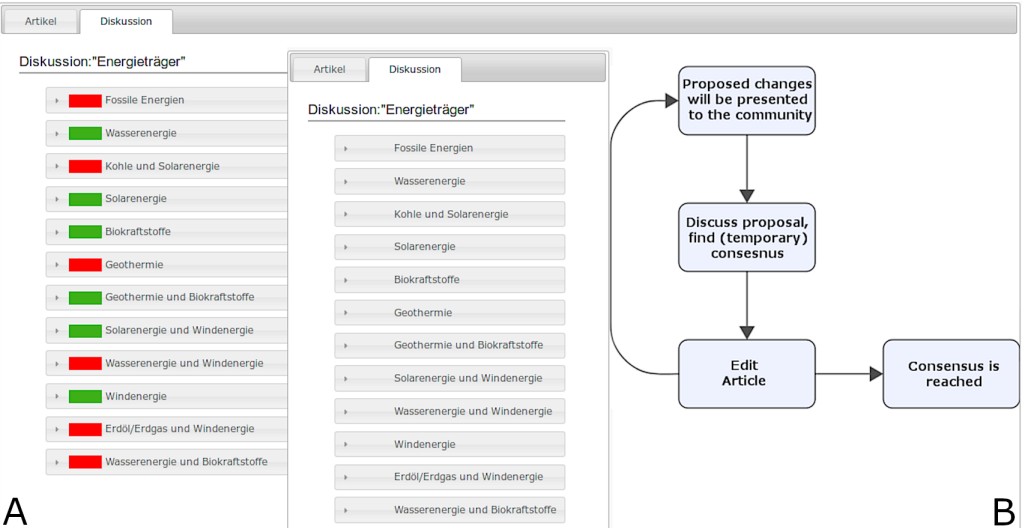

A        B

**Figure 1** (A) Representation of the implicit guidance wiki with controversy highlights (red indicators = unresolved controversies, green indicators = resolved controversies). (B) Representation of the explicit guidance wiki with the DDR collaboration script's core stages.

the DDR collaboration script was designed for discussion and deliberation of proposed changes to the wiki article, it was necessary to simulate this step to a minimal degree for this individual study. It was decided to shorten the presented discussions for this group by one reply of a previous discussant and adapt it as boilerplate text for a simulated bogus discussant. If a study participant in the script group decided to reply to a self-selected discussion, a pre-selection of decisions was presented with three options: (1) "I agree with discussant A", (2) "I agree with discussant B" and (3) "I agree with neither A or B / Both replies are equally valid to me." Depending on the user selection and the discussion status, one of three pre-defined bogus discussant replies was presented.

## Procedure

The experiment was conducted in an individual setup with up to four participants at the same time, separated by divider panels. After participants were individually briefed with written instructions on the computer screen and had given consent to participate in the study, they were first asked a few basic socio-demographics. This page also included assessments of interest in and prior knowledge of the study's subject matter, forms of energy. Participants completed all tasks of article editing and contributing to discussions individually in their own private wiki instances. This was followed by a short mandatory introduction to the self-developed wiki environment. Participants were asked to click through a mock-up environment with *lorem ipsum* texts to familiarize themselves with the general wiki structure. In addition to the general orientation in a wiki, this tutorial phase also served to familiarize them with the specific additions that were added to the experimental wikis to ensure that participants have a common ground about their wiki environment's mechanics. The group with controversy highlights for implicit

guidance received explanations about the meanings of the red (unresolved controversies) and green (resolved controversies) indicators. The group with the DDR collaboration script for explicit guidance received step-by-step text instructions about the core stages. As a reminder of the script's essential steps, they also received a permanently visible representation of the core stages as a flow chart.

Both groups had the same task of contributing to an initial Wikipedia-like base article about different forms of energy. Part of their wiki contribution task was also to participate in up to three discussions that accompanied the article. Participants received the information that the discussions contain sufficient arguments and evidence to enrich the original article. No other supplemental material regarding the subject matter was provided elsewhere. Participants did not receive further instructions on how to start their wiki task (e.g., reading the article or any discussion first) or what kind of reply they should make to a self-selected discussion. They were free to choose how to initiate their wiki experience. In the experiment's main stage participants had a loose total time limit of 21 min for finishing all article edits and discussion replies. For logging purposes, it was divided into three phases of 7 min for wiki contributions that included participation in a discussion and performance of an article edit. When the time for a contribution phase was up, the environment automatically prompted them to finish their contributions in the wiki and proceed further. Followed by the wiki contribution stage, the questionnaires to determine the individual levels of *Need for Cognitive Closure* (16-NCCS) and *epistemic curiosity* (ECS) were presented. After filling out these questionnaires, participants had to answer a multiple-choice test about the study's contents (cf. Fig. 2). As an additional manipulation check, participants were asked to sum up briefly in open text fields why they have selected certain discussions to comment on and what led to the final decisions for the resulting article edits. Finally, to gain insights about how participants rate the additions made to the wikis they were asked to fill out the *User Experience Questionnaire* (UEQ).

## Variables and measurements

The main independent factor of this study was the kind of additional wiki guidance, whether it was implicit with visual cues or explicit with a collaboration script suggesting a specific order of events. Another variable that also served for variance analytic purposes as a second independent factor was the individual Need for Cognitive Closure (NCC). According to the original literature on the scale 16-NCCS that has been deployed in this study, post-hoc median splits were used for classifying participants as low or high Need for Cognitive Closure (*Schlink & Walther, 2007*). Median splits should only be used with caution due to false estimations, lower power and spurious statistical significance (*Maxwell & Delaney, 1993*; *MacCallum et al., 2002*). Thus, the full continuous data spectrum of the Need for Cognitive Closure scale has been used within hierarchical linear regression models, in addition to the questionnaire's original evaluation protocol. On the other hand, median splits can also be useful to suggest clear recommendations to participants and to use these categories to build simple adaptive learning environments for different types of students. Thus, the following analytical procedure for process and

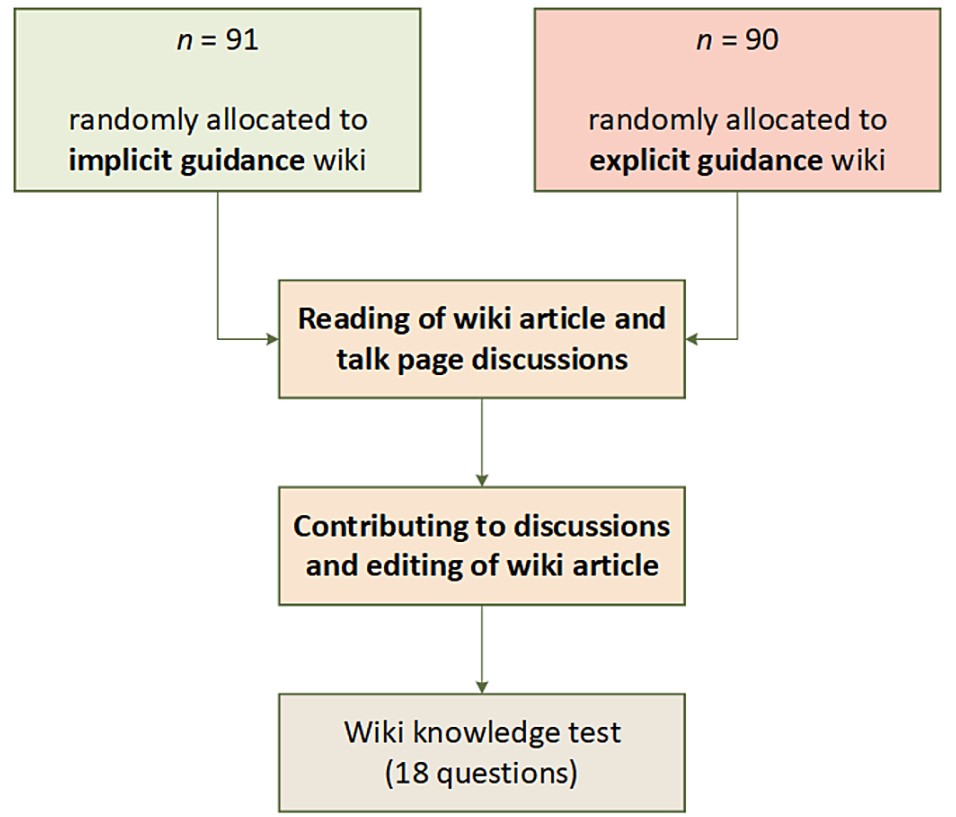

**Figure 2  Workflow diagram of the overall study procedure for the experiment with its central stages.**

outcome variables encompasses (1) analyses with dichotomised variables as well as (2) analyses with the entire metric data spectrum of the Need for Cognitive Closure scale.

### Measuring process variables

Both self-developed wiki environments were designed to record participants' selection behaviour by measuring individual clicks on the article and discussion tabs as well as on individual discussion thread titles. On the talk page, clicking on a title was necessary to select and expand a thread and thus unveil its contents (*topic selection*). By design only one topic could be open for reading at a time and had to be collapsed by clicking again before proceeding to the next topic of interest. For further processing click counts that triggered only the expanding/opening events were recorded in the log. Furthermore, the environment recorded the times of events such as time spent on the article page, time spent on individual discussions and time spent to write a reply to a topic (*topic contribution time*). Discussions' reading times were measured by calculating the differences between thread opening and closing times. If a topic was opened and closed more than once, the environment also recorded cumulative reading times for each discussion thread. Correspondingly, for the article page the environment's logging system also recorded the overall time spent with the article. Participants had to edit and reply three times during

the study and the system recorded the number of replies to each topic and which kind of controversy it was (*topic reply frequency*).

### Measuring learning success

To measure individual learning success about the study's subject matter, a post-experimental multiple-choice knowledge test was developed. Such tests are still widely used to quantify learning results in collaborative and individual settings (*Kent, Laslo & Rafaeli, 2016*). In total, the test comprised 18 questions about different forms of energy, such as different types of renewable energy sources (e.g., "What is the efficiency of water?", "What are the negative effects of wind turbines?"). Six of these questions were designed to be answerable with only the information provided in the original base article. Therefore, they were practically solvable without having read any of the discussion threads. The remaining 12 questions were constructed in a way that exactly one question covered one of the controversial discussion topics. Every multiple-choice question had four answering options comprised of up to three distractors and at least one attractor. The test's overall sum of correct answer options was used as a general indication for individual learning success about the study's subject matter. The theoretical maximum score a student could reach was 32.

### Measuring further potential influences

Individual epistemic curiosity was measured with the Epistemic Curiosity Scale (*Renner, 2006*). This validated questionnaire with a total of 10 items measures the two dimensions, diversive exploration and specific curiosity. Each of the two subscales consists of five statements (e.g., 'When I learn something new, I like to learn even more about it.') and had to be rated on a 4-point scale ranging from "fully disagree" to "fully agree". In addition to that, participants were also asked to rate their further experience with wikis in a short self-developed questionnaire with six items on a 4-point scale ranging from "not correct at all" to "fully correct". Items covered questions regarding passive use of and active participation in wikis.

### Measuring user experience

To measure the acceptance of the two guidance implementations which have been implemented in the experimental wikis, the *User Experience Questionnaire* (UEQ) was used (*Laugwitz, Held & Schrepp, 2008*). This questionnaire consists of 26 items measuring the perceived usefulness of user interfaces and their implementations. The UEQ does not provide a total score of the user experience, instead the construct is made up of six dimensions that are considered individually. Overall, values smaller than $M = -0.8$ are considered negative evaluations, values between $-0.8 < M < 0.8$ as neutral and values greater than $M = 0.8$ as positive evaluations of a tool on the respective dimension.

### Remarks on statistical analyses

For the following analyses, equivalence hypothesis tests have been used with the TOST (two one-sided $t$-Tests) procedure. This testing procedure is very useful when a null or very small effect is expected and has in general greater statistical power for gathering

evidence for or against the absence of effects. It applies two directional one-sided $t$-tests against a priori specified lower and upper effect size or raw score bounds. Simultaneously, a classic NHST (Null Hypothesis Significance Testing) $t$-test of differences from a null effect is applied. Within the TOST framework, the use of confidence intervals is more prevalent than $p$-values. Highly simplified, 95% CIs including zero correspond to non-significant $p$-values, whereas intervals excluding zero correspond to significant results (*Lakens, 2017a*). Equivalence hypothesis tests were performed with the R package *TOSTER* (version 0.3.3) by *Lakens (2017b)*. The underlying two one-sided $t$-test (TOST) procedure requires to determine a priori the smallest effect size of interest (SESOI) for the specification of equivalence bounds (*Hauck & Anderson, 1984*; *Schuirmann, 1987*). Applying the TOST procedure, in conjunction with a $t$-test of differences within the null hypothesis significance testing (NHST) framework, can yield four possible outcomes for an effect: (a) statistically equivalent and not different, (b) not equivalent and statistically different, (c) statistically equivalent and different, and (d) not equivalent and not different.

In the paragraphs regarding the analyses of interaction patterns, at first a $2 \times 2$ MANCOVA was used with the guidance type (implicit vs explicit) as group factor and the median split NCC (low vs high) as second factor. The dependant variables were the selection and reading behaviours of resolved and unresolved discussion topics. Participants' self-assessed subjective prior knowledge was used as covariate in all the following analyses because substantial differences between the experimental groups have been identified, $t(173.08) = 2.37$, $p = .019$, $d = 0.35$. As subsequent analyses, hierarchical linear regressions were used to account for the full continuous data spectrum of the Need for Cognitive Closure scale. For the analysis of the reply frequency as dependant variable, a $2 \times 2$ ANCOVA has been used instead of a MANCOVA.

# RESULTS

## Testing equivalence between guidance groups
### Topic selection
Participants in the implicit guidance wiki selected on average $M = 4.21$ ($SD = 2.75$) resolved topics in comparison to $M = 4.60$ ($SD = 2.86$) resolved topic selections in the explicit guidance wiki. The mean selection difference of 0.39 for resolved topics is equivalent in a 90% CI $[-1.17–0.21]$ and not significant in a 95% CI $[-1.30–0.34]$ within raw equivalence bounds of $-1.40$ and $1.40$ resolved topic selections. Unresolved discussion topics have been selected on average $M = 4.84$ ($SD = 3.88$) times in the implicit guidance wiki and $M = 4.24$ ($SD = 3.21$) in the explicit guidance wiki. The mean difference of 0.60 topic selections between the guided wiki groups is equivalent in a 90% CI $[-0.28–1.47]$ and not significant in a 95% CI $[-0.45–1.64]$ within raw equivalence bounds of $-1.78$ and $1.78$ unresolved topic selections.

### Topic reply frequency
Participants in the wiki with implicit guidance replied on average $M = 1.42$ ($SD = 0.92$) times to resolved controversies. In the explicit DDR script wiki, they replied $M = 1.80$

$(SD = 0.74)$ times to these controversial discussion topics. The mean difference in reply frequency between groups of 0.38 replies is not equivalent in a 90% CI [−0.58−−0.18] and statistically significant in a 95% CI [−0.63, −0.14] within raw equivalence bounds of −0.42 and 0.42. Since the number of total discussion replies was fixed to $n = 3$, analyses of the reply frequency to unresolved controversies would be completely redundant. Subsumed, the results in this and the previous paragraph provide evidence for hypothesis H1a where we assumed equivalence of implicit and explicit guidance wikis regarding the first set of process variables (*topic selection* and *topic reply frequency*).

### Topic contribution times

In the implicit guidance wiki participants spent on average $M = 386.46$ $(SD = 246.55)$ seconds on contributions to resolved controversial topics, whereas in the explicit guidance wiki $M = 481.11$ $(SD = 270.70)$ seconds were spent on contributing to these controversies. This mean contribution time difference of 94.64 seconds between both groups is not equivalent in a 90% CI [−158.30, −30.98] and also statistically significant in a 95% CI [−170.62, −18.67] within raw equivalence bounds of −129.45 and 129.45 s. Regarding unresolved controversial topics, participants in the implicit guidance group spent $M = 436.21$ $(SD = 298.56)$ seconds on contributions. In the explicit guidance wiki they spent $M = 341.37$ $(SD = 228.61)$ seconds on contributing to unresolved controversies. The mean difference in contribution times for unresolved controversial topics of 94.84 seconds is not equivalent in a 90% CI [29.51−160.17] and also statistically significant in a 95% CI [16.86−172.82] within raw equivalence bounds of −132.95 and 132.95 s. The second part of the analysis regarding unresolved controversies contradicts our assumption in hypothesis H1b that explicit guidance leads to generally longer contribution times.

### Knowledge test

In the knowledge test, participants in the implicit controversy highlight group scored on average $M = 15.84$ $(SD = 3.43)$ in comparison to an average score of $M = 15.47$ $(SD = 3.60)$ in the explicit scripting group. A mean test score difference between both guidance groups of 0.37 points is equivalent in a 90% CI [−0.50−1.23] and not significant in a 95% CI [−0.66, 1.40] within raw equivalence bounds of −1.76 and 1.76 points (Fig. 3). This result supports hypothesis H1c where we expected equivalence in learning outcomes on a group level, meaning that no guidance type is superior per se.

## Testing interactions of guidance and NCC
### Topic selection

In the MANCOVA model, the interaction effect of the experimental groups and the NCC level on topic selection behaviour was very small to virtually non-existent, $\lambda = .99$, $F(2, 175) = 0.83$, $p = .440$, $\eta_p^2 < .01$, 90% CI [.00−.04]. The follow-up regression analysis with the full continuous NCC data showed small effects for the interaction with the grouping variable on the selection of resolved topics with $b = 0.02$ $(SE = 0.04)$, $t(176) = 0.46$. This first analysis of time spent with resolved controversies topics supports hypothesis H1b, whereas t $p = .644$ within a small total effect model, $F(4, 176) = 0.72$, $p = .583$, $R^2 = .02$, 90% CI [.00−.04]. A similar pattern with small effects was found for the selection of
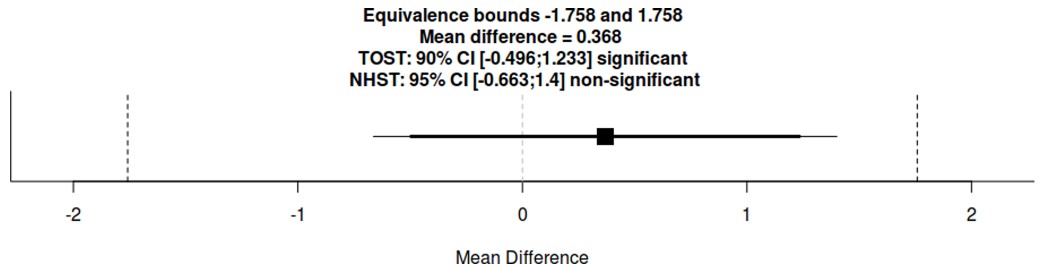

**Figure 3** Equivalence hypothesis test using the two one-sided *t*-Tests (TOST) procedure between experimental groups on mean test scores.

unresolved topics with $b = -0.02$ ($SE = 0.05$), $t(176) = -0.47$, $p = .640$ in the total effect model, $F(4, 176) = 1.05$, $p = .385$, $R^2 = .02$, 90% CI [.00–.05].

### Topic reply frequency

Due to the limitation of exactly $n = 3$ replies, instead of a MANCOVA on two reply variables a $2 \times 2$ ANCOVA has been used for analysing the interaction effect of guidance and NCC on resolved topic reply frequency. The effect with the dichotomized NCC was very small, $F(2, 175) = 1.91$, $p = .169$, $\eta_p^2 = .01$, 90% CI [.00–.06]. The regression model with the continuous NCC showed virtually the same pattern with a small effect with $b = 0.02$ ($SE = 0.01$), $t(176) = 1.32$, $p = .188$ within a rather large total effect model, $F(4, 176) = 6.24$, $p < .001$, $R^2 = .12$, 90% CI [.04–.19].

### Topic contribution times

For contribution times, the MANCOVA model showed an extremely small effect for the interaction of guidance type and the NCC, $\lambda = .99$, $F(2, 175) = 1.10$, $p = .335$, $\eta_p^2 < .01$, 90% CI [.00–.04]. Follow-up regressions with the NCC as continuous variable suggest some influence of the NCC interaction on the resolved topic contribution time with $b = 4.58$ ($SE = 3.61$), $t(176) = 1.27$, $p = .206$ within a medium-sized total effect model, $F(4, 176) = 3.03$, $p = .019$, $R^2 = .06$, 90% CI [.01–.11]. A much weaker effect was found for the regression on unresolved topic contributions time of $b = 1.12$ ($SE = 3.70$), $t(176) = 0.30$, $p = .762$ that was also in a medium-sized total effect model, $F(4, 176) = 3.07$, $p = .018$, $R^2 = .07$, 90% CI [.01–.12]. Hypotheses H2a and H2b are not supported by the data. The raw data of the process variables suggests that prior knowledge is primarily responsible for differences in replying frequency and contribution times for both resolved and unresolved controversial discussions.

### Knowledge test

Regarding the potential effects of the guidance type and the NCC level on the learning outcome, with a $2 \times 2$ ANCOVA a statistically significant small to moderate effect was found, $F(2, 175) = 5.30$, $p = .023$, $\eta_p^2 = .03$, 90% CI [.01–.11]. The Shapiro–Wilk test for normality suggests that the residuals in both experimental groups were close to normally distributed for (1) the knowledge test scores with $W = 0.98$, $p = .153$ (implicit), $W = 0.98$, $p = .150$ (explicit) and (2) for the NCC with $W = 0.99$, $p = .661$ (implicit),

$W = 0.98, p = .275$ (explicit). Using the full NCC data spectrum with a hierarchical linear regression model, the effect of the interaction itself was weakened, $b = -0.03$ ($SE = 0.05$), $t(176) = -0.67$, $p = .502$ within a moderate total effect model, $F(4, 176) = 3.20$, $p = .014$, $R^2 = .07$, 90% CI [.01–.12]. Both analyses provide evidence for the anticipated directional effects in hypothesis H2c, when controlling for prior knowledge differences. Figure 4 shows the interaction diagrams according to the conducted variance and regression analyses.

## Further measurements
### Testing equivalence between NCC levels
The following analyses on the process and outcome variables with the NCC as a dichotomised grouping factor are exploratory and were not part of our main research questions and hypotheses. We have performed them systematically in the same manner as we have done in the previous subsection with the main grouping factor (*guidance type*). We were interested to explore if there are any meaningful any differences in the dependent variables when the NCC is used as a main grouping factor.

*Topic selection.* Low NCC participants selected on average $M = 4.39$ ($SD = 2.59$) resolved topics in comparison to $M = 4.51$ ($SD = 3.04$) resolved topic selections of high NCC participants. The mean selection difference of 0.11 for resolved topics is equivalent in a 90% CI [−0.81–0.58] and not significant in a 95% CI [−0.94–0.72] within raw equivalence bounds of −1.41 and 1.41 resolved topic selections. Unresolved discussion topics have been selected on average $M = 4.35$ ($SD = 2.58$) times by low NCC participants and $M = 4.74$ ($SD = 4.36$) by high NCC participants. The mean difference of 0.39 topic selections between the guided wiki groups is equivalent in a 90% CI [−1.28–0.49] and not significant in a 95% CI [−1.45, 0.66] within raw equivalence bounds of −1.79 and 1.79 unresolved topic selections.

*Topic reply frequency.* Low NCC participants replied $M = 1.65$ ($SD = 0.86$) times on average to resolved controversies. High NCC participants replied $M = 1.56$ ($SD = 0.85$) times to these controversial discussion topics. The mean difference in reply frequency between groups of 0.09 replies is equivalent in a 90% CI [−0.12–0.03] and not significant in a 95% CI [−0.16–0.34] within raw equivalence bounds of −0.43 and 0.43. Since the number of total discussion replies was fixed to $n = 3$, analyses of the reply frequency to unresolved controversies would be completely redundant.

*Topic contribution times.* Low NCC participants spent on average $M = 427.15$ ($SD = 245.71$) seconds on contributions to resolved controversial topics, whereas high NCC participants spent $M = 440.11$ ($SD = 279.94$) s on contributing to these controversies. This mean contribution time difference of 12.96 s between both groups is equivalent in a 90% CI [−77.78–51.87] and not significant in a 95% CI [−90.33–64.41] within raw equivalence bounds of −131.69 and 131.69 s. Regarding unresolved controversial topics, low NCC participants spent an average time of $M = 409.79$ ($SD = 298.79$) seconds on contributions. High NCC participants spent $M = 367.61$ ($SD = 235.40$) seconds on

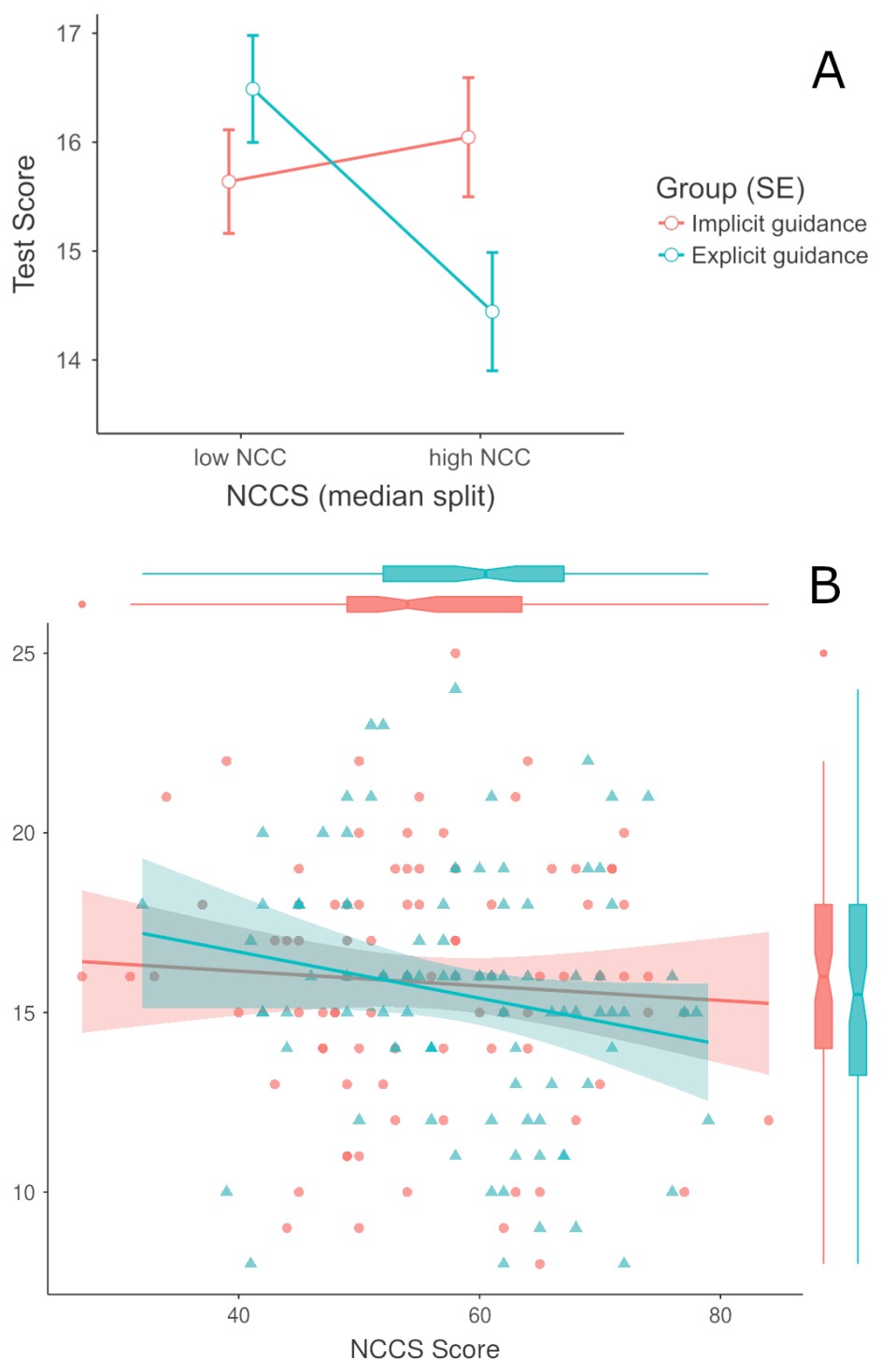

**Figure 4  Interaction diagrams of the guidance type with Need for Cognitive Closure (NCC) on knowledge test scores.** (A) The interaction is visualised as in a 2 × 2 ANOVA with the dichotomised (median split) NCC. (B) The linear regression slopes are visualised with smoothed standard error areas for the continuous NCC spectrum.

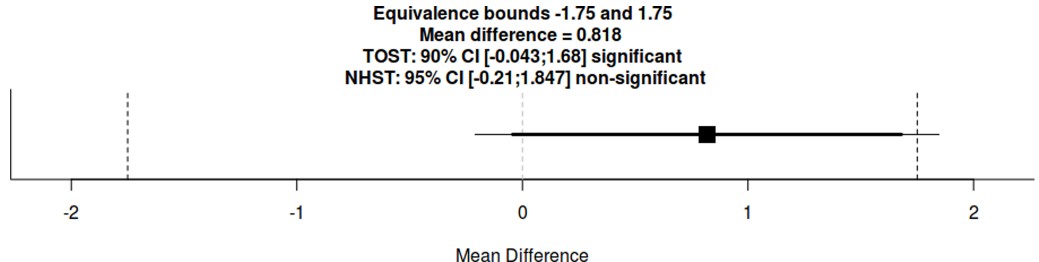

**Figure 5** Equivalence hypothesis test using the two one-sided $t$-tests (TOST) procedure between low and high NCC levels on mean test scores.

contributing to unresolved controversies. The mean difference in contribution times for unresolved controversial topics of 42.17 s is equivalent in a 90% CI $[-23.83–108.18]$ and not significant in a 95% CI $[-36.61–120.96]$ within raw equivalence bounds of $-134.49$ and 134.49 s.

*Knowledge test.* In the knowledge test, low NCC participants scored on average $M = 16.05$ ($SD = 3.29$) in comparison to an average score of $M = 15.24$ ($SD = 3.70$) of high NCC participants. A mean test score difference between both guidance groups of 0.82 points is equivalent in a 90% CI $[-0.04, 1.68]$ and not significant in a 95% CI $[-0.21–1.85]$ within raw equivalence bounds of $-1.75$ and 1.75 points (Fig. 5).

### Epistemic curiosity

Epistemic Curiosity was additionally explored as a construct that is related to the Need for Cognitive Closure. There was a small positive direct effect on the knowledge test scores, $b = 0.09$ ($SE = 0.06$), $t(176) = 1.54$, $p = .124$. Participants scoring high on Epistemic Curiosity perform minimally better in the knowledge test. The overall effect model in the hierarchical linear regression was medium-sized, $F(4, 176) = 4.65$, $p = .004$, $R^2 = .07$, 90% CI $[.01–.13]$. A relatively large portion of the effect is due to differences in prior knowledge. The data of analysing the interaction of Epistemic Curiosity and guidance suggests a small positive effect on learning outcomes, $b = 0.15$ ($SE = 0.11$), $t(176) = 1.40$, $p = .164$. Participants scoring low on the used curiosity scale perform slightly better with controversy awareness highlights for implicit guidance. In contrast to that, participants on the higher end of the curiosity scale perform slightly better with the DDR script for explicit guidance. The overall effect model of the interaction hierarchical was medium-sized, $F(4, 176) = 3.99$, $p = .004$, $R^2 = .08$, 90% CI $[.02–.14]$. As before, some portion of the effect in the overall model is due to prior knowledge.

### User experience

Overall, participants in the group with controversy awareness for implicit guidance rated the status highlights in five out of six UEQ dimensions higher than the collaboration script group for explicit guidance rated the DDR script. The largest difference on all six UEQ dimensions between both groups was on rating the *Efficiency* of their respective wiki environment, $t(179) = 6.44$, $p < .001$, $d = 0.96$. Students in the implicitly guided

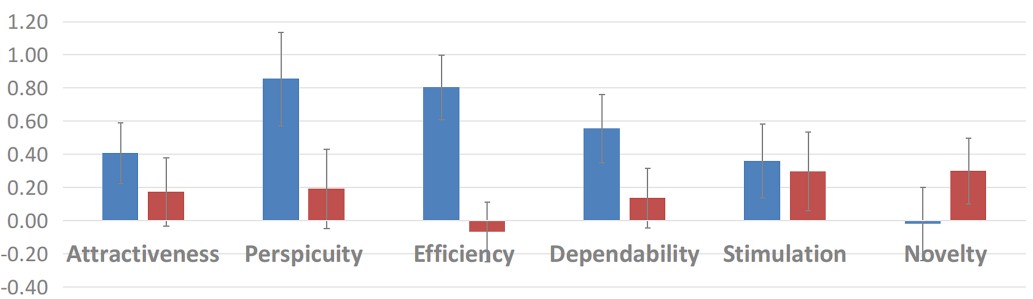

**Figure 6** **Average ratings on the dimensions of the User Experience Questionnaire.** Blue bars: controversy awareness highlights (implicit guidance); Red bars: collaboration script DDR (explicit guidance).

group rated their controversy status visualisations as rather positive in terms of efficiency whereas the explicitly guided experimental group gave even negative scores on the efficiency scale for the collaboration script. The only dimension where the explicitly guided group rated their wiki modification higher was the aspect of *Novelty*, $t(179) = -2.11, p = .037, d = 0.31$. See Fig. 6 for the detailed comparisons of all six UEQ dimensions between the experimental groups.

## DISCUSSION

We have already shown in previous studies that implicit and explicit guidance for educational wikis can influence behaviours of learners in a meaningful way. Implicit guidance aiming at controversy awareness with dedicated highlights can lead to a more focused selection of relevant content-related topics. Explicit guidance with a discussion-centric collaboration script, such as the DDR script for *deliberation, discussion and revision*, can lead to more meaningful a-priori discussion of proposed article changes (*Heimbuch & Bodemer, 2015a*; *Heimbuch & Bodemer, 2017*). But in those previous research attempts individual differences of learning-related variables were mostly measured as by-products of potential interest. Therefore, we laid the focus of this experimental study on one specific variable that had been previously identified as potentially relevant for learning scenarios where students are confronted with ambiguous information, namely the Need for Cognitive Closure (*Webster & Kruglanski, 1994*).

At first, we analysed the equivalence of implicit and explicit guidance methods in the overall sample. With regard to learning outcomes, there is no convincing evidence that one method in its own is superior per se. Our results show that participants do perform equivalently in the knowledge test, regardless of the guidance type (H1c). There was a slightly different pattern when we analysed the underlying process variables. Students showed equivalent behaviour in the topic selection process with no clear preferences (H1a). But groups differed significantly in their contribution times and replying behaviour (H1b). Students with implicit guidance spent more time for their wiki contributions when they were replying to unresolved controversies. With explicit guidance, students were more likely to reply to resolved controversies and invested more time for their contributions to the discussions. We looked into the comments

students could optionally provide at the end of the study to find some indications for these differences we did not expect. It seems the *red* highlight was a strong attractor for some students in the implicit guidance group. At the same time it was more difficult for these students to formulate an appropriate reply to controversial discussion that is still in an unresolved state. However, the full qualitative content analyses of this study are not yet complete at the time of writing (July 2018). As complementary analyses, we analysed the equivalence of students regarding their individual Need for Cognitive Closure (NCC), dichotomised in low and high levels (*Schlink & Walther, 2007*; *Schlink, 2009*). There were no meaningful differences between participants in the suggested categorised levels of the NCC on any of the measured process variables or in the learning outcome. This equivalence can be expected in such a setup with a time-constrained task, since a low NCC is not per se better than a high NCC in every situation. It depends not only on the individual disposition, there is also a strong situational factor (*Webster & Kruglanski, 1994*; *Kruglanski & Webster, 1996*). Thus, we were much more interested in the interaction of the NCC with different guidance types for computer-supported collaborative learning with wikis.

The interactions of the discussed guidance types and the Need for Cognitive Closure (NCC) were analysed with (M)ANCOVAs and hierarchical linear regressions. For all process variables, we found identical patterns with no meaningful differences in either statistical model (H2a, H2b). Deeper exploratory analyses of the data suggest that group differences in prior knowledge about the study's subject matter were crucial to differences in selection and replying behaviour. But when we analysed the knowledge test as learning outcome and controlling for prior knowledge differences, the interaction of guidance type and the NCC was following the anticipated pattern (H2c). Participants who scored relatively high on the NCC scale performed better in the knowledge test when they were made aware of controversies with implicit guidance in the wiki. This pattern reversed for low NCC students. They scored better in the explicit guidance wiki with the discussion-centric collaboration script. When we used a statistically more powerful framework, such as the proposed regression model, the interaction effect is much weaker. However, the general pattern of the interaction remains similar to the ANCOVA analysis with dichotomisation. Although the differences in raw test scores of approximately 1 to 1.5 are descriptively small on a maximum range of 0 to 32, they can still be meaningful. Even such a difference could be relevant for the next highest (or lowest) grade or can be a decision between passing and failing an assessment. Persons with a high NCC have the desire to resolve ambiguity as quickly as possible and care less for the best possible solution. Thus, they tend to rely on simpler heuristics to select and furthermore process information (*Dijksterhuis et al., 1996*; *Webster & Kruglanski, 1994*). Controversy awareness highlights for implicit guidance provide high NCC persons exactly this, a quick and easily accessible measure for assessment and decisions. In contrast to that, low NCC persons enjoy elaboration in discussions. They prefer to resolve ambiguity by finding better solutions than just the easiest solution available (*Dreu, Koole & Oldersma, 1999*; *Schlink, 2009*). Accordingly, a discussion-centric collaboration script like DDR that proposes the participation in discussion is better suited for them. Implicit and explicit guidance

measures in socio-technical learning environments can have immediate impacts on learning-related processes and learning outcomes. With this study, we showed that direct effects are not necessarily the case. Learning-related variables, specific to the requirements of an assessment, should also be considered. Thus, instructors and designers of learning environments can provide a more suitable learning experience that is more tailored to the individual prerequisites of learners.

Finally, we would like to note that students rate especially the implicit guidance implementation with controversy awareness highlights relatively positive in terms of user experience. The proposed highlights are easily understood and their use is generally perceived as efficient. Only in the dimension of *Novelty* the explicit guidance with the collaboration script DDR was rated higher. This is likely due to the fact that most users are familiar with other visualisations of similar information, especially from the context of social media (e.g., Facebook, Twitter, Google +). Contrary to this, it is most likely that many students have been confronted with collaboration scripts for the first time within the course of this experimental study.

### Limitations and future research

As with most research conducted in higher education, this experiment was conducted with a very specific student sample at a single university (University of Duisburg–Essen, Germany) and mainly from a fairly unique degree programme, namely the Applied Cognitive and Media Sciences—an interdisciplinary program that combines the subjects of psychology and computer science (https://www.uni-due.de/komedia). These students tend to be rather inclined to digital media. Therefore, we also believe that it would be very likely that the results are replicable within other populations of more than average technology-affine students. It would be also very interesting to investigate the effects in populations that are less proficient with computers and socio-technical systems, because even in highly developed countries like Germany where this study was conducted, there are huge differences between students of various backgrounds (*Fraillon et al., 2014*). We would like to investigate if similar patterns as reported in this research article would emerge. Furthermore, with regard to the available study data we are still interested in investigating potential differences of the contribution quality of both wiki guides presented here. The qualitative data analyses are still ongoing and planned to be finished at the end of the third quarter 2018.

### CONCLUSIONS

Based on our findings, we conclude that students in wiki-based learning can be supported more effectively with specific guidance measures when also relevant individual differences are considered. In an environment that is suited for the co-construction of socially shared artefacts and socio-cognitive conflicts to occur, we suggest that students receive supportive guidance that fits them. On the one hand, if the personal Need for Cognitive Closure is low and students enjoy dealing with ambiguous information, they could benefit most from explicit guidance through collaborations scripts that encourage them to discuss controversial points of view and deliberate how to further proceed in the wiki.

On the other hand, high closure students could benefit more from minimal obtrusive implicit guidance that signals them potentially relevant information in wiki discussions and lets them choose to self-regulate what kind of controversial discussions they like to deal with. It is also worth noting that when either the guidance types or the Need for Cognitive Closure were analysed in isolation, equivalence in terms of learning success has been found. That means neither group is per se better than the other. In conclusion, this experiment provides some evidence that individual differences in variables such as the Need for Cognitive Closure should be considered in designing and deploying learning environments where opposing evidence and controversies play a key role.

### Funding
The authors received no funding for this work.

### Competing Interests
The authors declare there are no competing interests.

### Author Contributions
- Sven Heimbuch conceived and designed the experiments, performed the experiments, analyzed the data, contributed reagents/materials/analysis tools, prepared figures and/or tables, authored or reviewed drafts of the paper, approved the final draft.
- Daniel Bodemer authored or reviewed drafts of the paper, approved the final draft.

### Human Ethics
The following information was supplied relating to ethical approvals (i.e., approving body and any reference numbers):

The University of Duisburg-Essen granted Ethical approval to carry out the study within its facilities (Ethical Application Ref: psychmeth_2015_WIKI_08).

### Data Availability
Heimbuch S. 2018. Exploring the interplay of scaffolding and the need for cognitive closure in wiki-based learning. http://osf.io/ut2aq.

### Supplemental Information
Supplemental information for this article can be found online at http://dx.doi.org/10.7717/peerj.5541#supplemental-information.

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
