# Peer review of "Interaction of guidance types and the Need for Cognitive Closure in wiki-based learning"

_PeerJ, doi:10.7717/peerj.5541_

## Round 0.1 · original submission · Minor Revisions

Please address all issues raised by your peer reviewers, while using discretion on resolving comments in the review that suggest Major Revisions, addressing all reviewer matters that make sense in helping you finalize the highest quality paper.

Minor issues, grammatical, exist and need resolved such as this opening statement:

"The prevalence of collaborative knowledge construction environments such as wikis play..."

The sentence subject is 'prevalence' which is singular, so it should read as below with an expected plural verb:

"The prevalence of collaborative knowledge construction environments such as wikis plays..."

Or better:

"The prevalence of collaborative knowledge construction environments, such as wikis, plays..."

But okay. Point is, with that comment, there are places throughout where this manuscript would benefit grammatically from a comma here or there or breakup of compound run-on sentences that read "...___ and ___ blahblahblah and ___ and __ blahblah..." as these occur rather frequently. Please resolve as many as seems sensible to address in your final proof for resubmission.

Finally, should this manuscript be using a style similar to APA style but not exact, then my comments hereafter may be moot. But if its intention is to follow APA style as revealed largely therein, then ampersands (&) belong in the ()s and not the word 'and' there so they should all look like (Heimbuch & Bodemer, 2018). That would be everywhere in the document; unless, of course, they are in the text correctly expressed as "...Heimbuch and Bodemer in 2018 said..." Ampersands also belong in the References list everywhere expected:

Heimbuch, S., & Bodemer, D. (2018). Blahblahblah.

Thank you for the opportunity to review your manuscript. I am looking forward to your submitted revision with noted minor changes.

·

Basic reporting

The introduction, method and results sections are very well written and are clear, while the discussion and conclusion are less clear and contain more errors, although still clear. I'd suggest that a native English speaker proof reads these two sections.

The abstract doesn't misrepresent the results of the paper, but could be clearer about the actual findings, which are (in my opinion) somewhat less interesting than might be assumed from the abstract, rather than simply hinting about 'effects'. e.g. a short list of how the guidance types affect user behaviour and how NCC interacts with 'learning outcomes', namely the test score (e.g. the sentence starting line 455).

Similarly, the non interactions of the study variables are ignored in the abstract. e.g. line 419, "In summary, the analyses suggest relatively small to virtual no effects on the interaction between guidance and the NCC on all the process variables" or the line 469.

Figure 2 is too small to read.

The abbreviation DDR is introduced on line 118, but isn't defined anywhere I could find.

On this topic, the language used is sometimes mixed, e.g. implicit vs. explicit guidance, highlights vs. script, icons vs. DDR, etc. Although it's pedantic, I think you should pick one way of referring to the two wikis and use it consistently throughout the introduction, results and conclusion.

Experimental design

I'm tempted to put this paper more in the area of Social Sciences, or the Humanities rather than clearly in the areas of Biological Sciences, Medical Sciences, or Health Sciences.

The design and motivation of the experiment is good and the study rigorous. The methods are clear.

Validity of the findings

The statistical analysis looks robust to me (not technical expert). I'd recommend that at least one of the reviewers be professional statistician.

One control which could be interesting is a wiki without any guidance, implicit or explicit, but I think the authors mention their previous work in this area. This could probably be made clearer.

Additional comments

The work is good and the introduction is very interesting and flows well.

I think you should be clearer when you're citing your own work. e.g. 'we have previously shown...' rather than, 'previous experimental studies have shown...'.

NCC is mentioned on line 50 of the introduction, but isn't described until later in the paper (line 105). I'd be happier if it was described the first time it is mentioned.

Why is this RQ5? what happened to RQ1-4?

I'd be interested in an evaluation of the quality of the final articles.

Sorry for the dumb question, but, why is the mean selection difference of 0.48 (line 319) not simply 4.60 - 4.21? e.g. 0.39?

Does Figure 5 show statistically significant differences? This could be clearer.

·

Basic reporting

1/ The title is not relevant.
- The "scaffolding" term is undefined in the article.
- The study design is not mentionned (laboratory experimental study).

2/ Background and research justification are unclear
Theorical concepts in introduction are exposed confusingly, and the relevancy of the research questions is not understandable. It is only at the beginning of the discussion (lines 452-460) that the real point of interrest of the study seams to appear...
We would be greateful to the authors:
- to expose clearly in introduction the fact that this study is a continuum with prior works. Please avoid citing oneself as any else author (l. 48-49)
- to recall main findings of their anterior studies and to explain more directly the relevance of this further investigation before exposing research questions and hypothesis (l. 119-130 and 141-147 are unexpected at this stage). Does the research question number "RQ5" indicates that this paper is only a part of a larger research framework ? If yes, please state. If no, please explain.
- to deepen the state of the art regarding to the study concern: are there other studies focused on NCC and collaborative learning guidance ?
- to avoid using concepts without defining them ("need for cognitive closure" appears at line 50 while it is only defined at line 103)
- to shorten long sentences, especially those referring to complex theorical knowledge (l. 43-45)
- to cite litterature accordingly to affirmations
-- l. 22 - Choy and Ng 2007: we expect a paper giving an idea of the prevalence of the use of collaborative learning tools in higher education, not a single experimentation report
-- l. 24 - Notari et al. 2016: we expect a descriptive paper about uses of wikis within the educational field, not a referrence to a method for constructing educationnal wikis. At least, indicate the pages in this book giving such a description.
-- l. 31-33: no citation. Referring to the athors' prior works ??
(...)
- to lighten the theorical background exposure when having no impact on the study's relevancy (l. 21 - 115)
- to give a definition for every acronyms (l. 118)

Experimental design

3/ There is a damaging lack of correspondance between the research questions (RQ), the methods and the results
The reader can not link findings to the RQ nor to the methods:
- the RQ terms 'individual selection', 'reply behaviour' and 'individual contribution' do not match the results terms 'topic selection', 'topic contribution time' and 'topic frequency' without explanation.
- the methods terms exposed in "Measuring process variables" (l. 273-283) do not match the result terms 'topic selection', 'topic contribution time' and 'topic frequency':
-- it seems that the results' "topic selection" correspond to the methods' "click on talk page titles" ? - please explain
-- to what correspond the results' "contribution times" in the methods ? article reading time ? talk threads reading time ? article edition time ? talk threads response edition time ?
-- to what correspond the results "reply frequency" ? article's chapters ? talk threads ?
- RQ, methods and results do not follow the same plan, which makes the cross-reading extremely difficult.
We would be greateful to the authors to clarify the terms used and to harmonize RQ/methods/results.

4/ The Methods needs clarifications
- how were recruited participants ? according to which criteria ?
- there is no description of the origin of participants recruited out of the university (N=13)
- there is no description of gender, age nor origin distribution within groups
- while using t-test based statistical tools, it would be welcome to test the normality of the distribution of NCC values (l. 248-261)
- statistical calculations upon the knowledge test are made without giving a description of it (l.177-179): we need at least the scale of the test to appreciate the 1 point minimal effect size of interrest.
- the participant's topic interrests/prior knowledge are presented without having exposed which is the topic concerned (l. 192)
- the "base wikipedia-like" article is mentionned (l.211) before it's description (l.229 and further)
- figure 1 is extremely simplified according to the real participant's process. It does not allow a better understanding.
- the "3 phases of 7 minutes" (l.217) need deeper description as they seem to impact the results understanding (what really mean "topic reply frequency" ?)
- !! this is an important point as it sustain all the study: the description of the wiki environment and the two types of guidance need to be improved (l.234-246 and figure 2)
-- a screenshot of the article would be welcome
-- avoid using undefined acronyms ("DDR") or familiar words ("bogus")
-- comparative screenshots of both groups with more explanations are needed:
--- in the implicit group: authors should legend flags colors meaning
--- how look like the same screen for the explicit group ? how and when appear explicit guidance messages on screen ? how does the participant's choice (1/2/3/(4?)) affects the page edition ?
--- the diagram on the right of the figure 2 is disconnected from the text description
- the dataset contains 'TimeResolved', 'TimeUnresolved', 'ReplyResolved', 'ReplyUnresolved','ArticleLength' and 'ArticleDifference' colums that do not match with methods description nor results terms. Authors should precise:
- what include 'TimeResolved' and 'TimeUnresolved' items: talk reading ? article reading? talk response ? article edition ?
- how were considered 'replies' contributions: only in talks? including the correspondant section in of the article ?
- why authors did not mention the articles' length in their paper while having measured it ? Please justify every step of your work, even abandonned branchs.

Validity of the findings

5/ The results exposure is unclear
- results should not include methods (l. 301-315, 385-394)
- as already pointed, results should follow same items than RQ and methods
- Figures 3 and 4 do not improve the understanding. Authors should report every measure (topic selection / contributon times/reply frequency/knowledge test) or none.
- The knowledge test interaction between guidance and NCC analysis do not report the effect of the covariate "prior knowledge" which is supposed to have been tested.
- Figure 5 shows the linear regression result with points dispersed on a 0-25 scale ordinate as the test score ranks 0-18: ?

6/ Lacks in the presentation of the findings (discussion/conclusion)
In Discussion, we would appreciate that authors discuss strength and weakness of their work, in particular:
- the fact that most participants are Applied Cognitive and Media Sciences students is a limitation in results extrapolation (which is developed in conclusion ... )
- the fact that the study design and methods were adapted to existing tools (implicit and explicit guidance processes) rather than having creating tools on purpose.
- these findings seems to be of some interrest to improve individually-fitted collaborative learning processes. However, the "real life" extrapolation of these findings should be discussed.
As already expressed, the validity of the findings is impaired by the lacks in methods. In particular, we need to better understand the correspondance between the results and the participants' tasks.
The Conclusion is actually a Discussion. We expect there a brief synthesis of main findings in order to replace them in the general context, eventually followed by an opening.

Additional comments

7/ The scope of this paper is at the fronteer of the cognitif sciences. It is more relevant to educational sciences & informatics.

Thanks to the authors for this work, and to the editorial team of PeerJ to trust my expertise.

Reviewer 3 ·

Basic reporting

This was a well structured and clearly articulated article. The literature was well described, although two relatively minor points are: Lines 111-113: Citation needed
Lines 113-115: You mention that there are “only a few studies” that address the concept but don’t provide them. Please clarify and/or provide a further sentence/s here to identify sources that do address it. The diagrams and figures were professionally produced. Hypotheses were clear with relevant results. (Note that there were quite a few typos in the article e.g. spelling of "conclusions" in the final section).

Experimental design

The research fits within the aims and scope of the journal and the rationale for the research is clearly articulated. The research design was relevant to the research questions which were meaningful to current questions relating to the design of wiki tools for collaborative educational purposes.

Validity of the findings

Conclusion is clear and supports the results reached.

---

## Round 0.2 · Minor Revisions

Authors should seriously consider the validity of the re-review, and provide an appropriate revision or fair argument for each valid claim raised by the reviewer, as to why they consider each suggestion valid and thereby newly included, or invalid and rejected. While the paper appears otherwise solid, it will benefit from this consideration at this time before rendered any final editorial decision.

·

Basic reporting

I would first congratulate the authors. The article has been significatively improved since the first version. Review comments have been seriously considered and revised accordingly. The global text thread is now well understandable, introduction, methods, results and discussion are coherent, and the justification of the research questions is finally clear.

Some minor comments/questions :

Experimental design

Methods :
- i would have liked to have a brief description of the 13 participants which were not students, and particularly : how are they distributed in the two groups ? (e.g. : would it represent a confusion variable?)

- "Regarding the normality assumption checks of t-Tests. We omitted these results since the equivalence analysis we use builds upon Welch’s and not Student’s t-Test.[...]"

My note aimed at helping you improving the validity of your study, especially since your main finding lays on a variance analysis. I had a look at your NCC's values: they actually seem normally distributed. But that hypothesis should be tested. I agree that the Welsch's correction of the Student's t-test improve the robustness regarding to variance gaps. However, it is not the case regarding to non parametric distributions. Such a weakness would dicrease in large samples, but it is not the case of this study. More critical for your findings: analysis of variances such as (M)ANCOVAs would make no sense in case of non parametric distribution.

Validity of the findings

Results :
- some RQ hypothesis are mentioned to be validated (l 362, 368). However, as results are not layed in the same order as hypothesis are given, and as hypothesis and results are multiple and slightly complex, the reader must go up and down several times through the text to well understand correspondances and meanings. Authors should refer more clearly to their RQ (e.g. «  Subsumed, the results regarding the process variables provide evidence for the hypothesis H1a about the equivalence of implicit and explicit guidance wikis. »). Doing this way systematically for every hypothesis would be greatfull.
- hypothesis H1b and Topic contribution times : how do you interprete the results ? They show significative differences between groups, but in opposite ways depending on the discussion threads quality (resolve/unresolved). However, « resolved/unresolved » was not a point of the RQ1 ?

Discussion :
- the discussion about the interaction of NCC & guindance mainly concern the hypothesis H2c. What about H2a and H2b, which seem not verified ?

Additional comments

After all, the conclusion is in accordance with the findings, and, moreover, the abstract resumes perfectly the whole study.

This is a quality paper which will be probably usefull in a broader scope than the single wiki-based learning.

Thanks

---

## Round 0.3 · accepted · Accept

The most recent revision is accepted with an understanding that annotated adjustments in the attached PDF are appropriately adopted by the authors prior to PeerJ publication, so the current manuscript must be finalized as per these annotations to ultimately fulfill this approval.

#